# Target-Site Mutations and Expression of *ALS* Gene Copies Vary According to *Echinochloa* Species

**DOI:** 10.3390/genes12111841

**Published:** 2021-11-22

**Authors:** Silvia Panozzo, Elisa Mascanzoni, Laura Scarabel, Andrea Milani, Giliardi Dalazen, Aldo J. Merotto, Patrick J. Tranel, Maurizio Sattin

**Affiliations:** 1Institute for Sustainable Plant Protection (IPSP-CNR), National Research Council of Italy, Viale dell’Università 16, 35020 Legnaro, Italy; laura.scarabel@cnr.it (L.S.); andrea.milani@ipsp.cnr.it (A.M.); maurizio.sattin@cnr.it (M.S.); 2University of Padova, 35020 Legnaro, Italy; elisa.mascanzoni@corteva.com; 3State University of Londrina, Londrina 86057-970, Brazil; giliardidalazen@gmail.com; 4Department of Crop Science, Federal University of Rio Grande do Sul, Porto Alegre 91540-000, Brazil; merotto@ufrgs.br; 5Department of Crop Sciences, University of Illinois, Urbana, IL 61801, USA; tranel@illinois.edu

**Keywords:** *Echinochloa* spp., barnyard grass, late watergrass, DNA barcoding, ALS inhibitors resistance, target-site resistance, Ala122Asn, Trp574Leu, *ALS* gene copies, *ALS* alleles expression

## Abstract

The sustainability of rice cropping systems is jeopardized by the large number and variety of populations of polyploid *Echinochloa* spp. resistant to ALS inhibitors. Better knowledge of the *Echinochloa* species present in Italian rice fields and the study of *ALS* genes involved in target-site resistance could significantly contribute to a better understanding of resistance evolution and management. Using a CAPS-*rbcL* molecular marker, two species, *E. crus-galli* (L.) P. Beauv. and *E. oryzicola* (Vasinger) Vasing., were identified as the most common species in rice in Italy. Mutations involved in ALS inhibitor resistance in the different species were identified and associated with the *ALS* homoeologs. The relative expression of the *ALS* gene copies was evaluated. Molecular characterization led to the identification of three *ALS* genes in *E. crus-galli* and two in *E. oryzicola*. The two species also carried different point mutations conferring resistance: Ala122Asn in *E. crus-galli* and Trp574Leu in *E. oryzicola*. Mutations were carried in the same gene copy (*ALS1*), which was significantly more expressed than the other copies (*ALS2* and *ALS3*) in both species. These results explain the high resistance level of these populations and why mutations in the other *ALS* copies are not involved in herbicide resistance.

## 1. Introduction

The genus *Echinochloa* includes about approximately 50 annual summer species, widespread in both tropical and temperate regions and in dry (e.g., maize and soybean) or water flooded soils (e.g., rice) [1]. Several classification keys have been proposed in the last century, but none of them have been able to adequately fulfill the task; *Echinochloa* species are often very difficult to distinguish due to wide intraspecific morphological and phenological variability [2]. A rough first classification can be done on the basis of macro-phenological differences, dividing *Echinochloa* species in two groups: the “red” (i.e., *E. crus-galli* (L.) P. Beav. and *E. hispidula* (Retz.) Nees) and “white” species (e.g., *E. oryzicola* (Vasinger) Vasing. and *E. oryzoides* (Ard.) Fritsch). *E. oryzicola* (late watergrass) and *E. crus-galli* (barnyard grass) are the most common members of the two categories in Italian paddy fields. *E. oryzicola* is the dominant and most persistent due to its complex survival strategy in flooded rice [3]. The species of both groups are annual with a C_4_ photosynthetic pathway [3,4] and show a great competitive advantage when they grow together with C_3_ crops.

*Echinochloa* spp. are weedy plants that have evolved resistance to several herbicide classes, among which acetolactate synthase (ALS) inhibitors are the most used for their control. The ALS-inhibiting herbicides are commonly used because of their high activity at low doses against a broad spectrum of weeds, the possibility of post-emergence grass weed control, the low mammalian toxicity and low costs [5]. Furthermore, the introduction of the Clearfield^®^ technology, i.e., imidazolinone-tolerant rice varieties, has further increased the use of these herbicides, reducing the diversity of sites of action used in rice crops [6]. Their highly specific target, along with their repetitive use on the same cropping system, have favored the selection of resistant populations. In most cases, the resistance to ALS inhibitors is due to point mutations in the *ALS* gene (i.e., target-site resistance, TSR), that change specific amino acids of the target protein (ALS), thus reducing its affinity for this class of herbicides [7]. So far, TSR to ALS inhibitors in *Echinochloa* involve three amino acid positions: Ala-122, with three allelic variants (Val [8], Thr [8] and Asn [9], in *E. crus-galli*), Pro-197, with two allelic variants (Ser and Leu, in *E. cru-galli* [10]), and Trp-574, with one allelic variant (Leu, in *E. crus-galli* [10,11] and *E. oryzicola* [12]).

On the basis of cytological evidence, *E. crus-galli* is an allohexaploid assumed to have derived from a hybridization between the allotetraploid *E. oryzicola* and an unknown diploid species [13]. In polyploid species a variable number of each gene can occur. Therefore, the genomic organization is more complex than that of diploid species [14,15]. *E. crus-galli* and *E. oryzicola* are allopolyploid species, which contain three and two homologous *ALS* genes, respectively [16], and each gene copy encoding a protein could evolve resistance-endowing mutations [7]. It is still unclear how many gene copies, and which specific copy/ies, are responsible for the evolution of herbicide resistance in *Echinochloa* and other polyploid species.

In this study, molecular analyses were performed on several *Echinochloa* spp. field populations to achieve a better taxonomic classification of “red” *E. crus-galli* and “white” *E. oryzicola* species using a CAPS Cleaved Amplified Polymorphic Sequence (CAPS) molecular marker based on the DNA barcoding method. The mutations endowing resistance to ALS inhibitors in the different species were detected and associated with the different *ALS* gene copies. Lastly, the relative expression of the *ALS* gene copies was evaluated to explain the high level of ALS inhibitor resistance in these polyploid species.

## 2. Materials and Methods

### 2.1. Plant Material

Seven *Echinochloa* spp. populations were analyzed: 16S, 44R, 45R, 46R, 95R, 100R and 161S (note: S = susceptible and R = resistant). The different populations were preliminarily classified using the Pignatti (1982) [17], Costea and Tardif (2002) [18] and Viggiani and Tabacchi (2017) [19] classification keys. All populations were collected in Italian rice fields (Appendix A), and tested for herbicide resistance through greenhouse assays following the protocol described in Panozzo et al. (2015) [20]. Leaf tissue from the survivors of the recommended field dose of penoxsulam (40.8 g a.i. ha^−1^) were sampled, frozen in liquid nitrogen and conserved at −80 °C until nucleic acids extraction.

### 2.2. DNA Barcoding and CAPS-rbcL

A DNA barcoding marker was established to assign the tested plants into the two classes of *Echinochloa* species (“red” or “white”). This was important to avoid misidentifications due to the high heterogeneity of field-collected populations, which can include plants from different species, and because correct identification of the plants was required for the molecular analyses.

A preliminary analysis using MEGA X^®^ software [21] was performed including several nucleotide sequences of *Echinochloa* spp. cpDNA (chloroplast DNA) gene *rbcL* [22,23] reported in GenBank database (www.ncbi.nlm.nih.gov/nucleotide accessed on 10 December 2021) and in the Barcode of Life Data Systems (BOLD, http://www.barcodinglife.org/ accessed on 1 September 2021, vouchered specimens only), as well as some *rbcL* sequences obtained from *Echinochloa* purified and classified accessions [24].

Genomic DNA (gDNA) was extracted from leaf tissue of five plants for each population using the CTAB method described in Doyle and Doyle (1987) [25]. Concentration and quality of gDNA was determined using a NanoDrop™ 2000 (ThermoFisher Scientific, Waltham, MA, USA). Each sample was diluted with ddH_2_O to reach a final concentration of 100 ng µL^−1^ and then stored at −20 °C until use.

PCR amplification was conducted using the GoTaq^®^ G2 Hot Start DNA Polymerase (Promega, Madison, WI, USA) in a 25 µL final volume mixture including 5 µL 5× Go Colorless GoTaq^®^ Reaction Buffer, 2.5 µL MgCl_2_ 25 mM, 0.5 µL dNTPs mix 10 mM, 1 µL of primer rbcL_F1 (5′-GCA GCA TTC CGA GTA ACT CCT CA-3′) [24] 10 µM and 1 µL of primer rbcL_R2 (5′-TTG GTG GAG GAA CTT TAG GAC ATC-3′) 10 µM, 0.2 µL GoTaq^®^ G2 Hot Start DNA Polymerase and 1 µL of gDNA. PCR reaction was conducted in a T1 Thermocycler (Biometra, Göttingen, Germany) using the following program: 2 min at 95 °C, 35 cycles with 30 s at 95 °C, 30 s at 60 °C, and 80 s at 72 °C, plus a final extension step at 72 °C for 5 min. Amplicons were purified using the NucleoSpin^®^ Gel and PCR Clean-Up Kit (Macherey-Nagel, Allentown, PA, USA) and sequenced by BMR genomics (Padova, Italy). Sequences were analyzed with Finch TV 1.4.0 software; consensus sequences were built using the SeqMan software included in the package DNASTAR^®^. MEGA X^®^ software was used to align the sequences and to build UPGMA dendrograms.

To quickly distinguish the “red” from “white” plants in a heterogeneous population avoiding longer and more expensive sequencing experiments, a chloroplast cpDNA-based Cleaved Amplified Polymorphic Sequence (CAPS) system was designed based on the results of the DNA barcoding marker [26]. Two non-species-specific primers were designed on the multi-species alignment of the *rbcL* gene sequences, namely CAPS-rbcl-F (5′-CAACTGTTTGGACTGATGGAC-3′) and CAPS-rbcl-R (5′-CGTAGATCCTCCAAACGTAGAGC-3′); only the *rbcL* sequence of “white” plants had a restriction site for the endonuclease TasI (AATT) (Appendix A).

The PCR of the CAPS assay was performed using the Go Taq^®^ G2 DNA Polymerase (Promega) in a 25 µL final volume mixture adding the required reagents. Amplification was conducted using the following program: DNA denaturation for 2 min at 95 °C, 25 cycles of 30 s at 95 °C, 30 s at 55 °C, and 20 s at 72 °C, and a final extension step of 5 min at 72 °C. PCR products were purified as described above and digestion with the endonuclease TasI (ThermoFisher Scientific, Waltham, MA, USA) was set as follows: purified PCR reaction mixture (5 µL), 10× Buffer B (1 µL) and TasI (1 µL) in a total volume of 16 µL. The reaction mix was incubated at 65 °C for 1 h and the inactivation of TasI was done by incubation at 80 °C for 20 min. EDTA 0.5M pH 8.0 (0.64 µL) was added to prepare the digested DNA for electrophoresis and the run was performed in a 2% agarose gel at 75 V for 90 min. Six to ten individual plants of each population were tested with the CAPS-rbcL to validate the method and then all plants used in the following experiments were preliminarily checked with the molecular marker.

### 2.3. Southern Blotting with ALS Gene

The plant material consisted of one *E. crus-galli* population (16S) susceptible to ALS inhibitors and three *E. oryzicola* populations, one susceptible (161S) and two resistant to ALS inhibitors (45R and 46R) collected in two sites far away from each other. This choice was made to determine whether the different number of copies of the gene could be related to the species and/or to the resistance status and/or to the site of collection.

Seeds of each population were chemically scarified [11] and placed in soil. When seedlings were at three-leaf stage, one gram of the youngest leaf tissue (pooled from 10 plants) per population was ground in liquid nitrogen using a mortar and pestle and gDNA was extracted as described in Section 2.2.

Digoxigenin (DIG)-labeled probe was generated by amplification of a conserved region (around B domain) of the *ALS* gene from gDNA of susceptible plants (population 16S) using PCR DIG Probe Synthesis Mix (Roche, Basel, Switzerland) and SB-F2 (5′-TGAGTTGGATCAGCAGAAGAG-3′) and SB-R2 (5′-AAGACCTTCACTGGGAGGTTC-3′) primers. GoTaq^®^ Flexi DNA Polymerase (Promega) was used for the amplification. To a final volume of 50 µL, the following reagents were added: 10 µL 5× Colorless GoTaq^®^ Flexi buffer, 2 µL MgCl_2_ solution 25 mM, 5 µL PCR DIG Probe Synthesis Mix (Roche), 2 µL of each primer 10 µM, 2.5 U of GoTaq^®^ Hot Start DNA Polymerase and 200 ng of gDNA. Amplification was conducted using the following program: 2 min at 95 °C, 30 cycles of 30 s at 95 °C, 30 s at 54 °C, and 40 s at 72 °C, and 5 min of final extension time at 72 °C. The probe was purified using the MinElute PCR purification kit (Qiagen, Hilden, Germany) and stored at −20 °C until the digestion with endonucleases.

Five, 6-nucleotide-recognizing endonucleases that did not cut the probe sequence were selected to perform Southern blotting analysis on the four *Echinochloa* populations. Three experiments were conducted: (1) gDNA of populations 16S and 45R was digested with two restriction endonucleases, HindIII (Invitrogen, Waltham, MA, USA) and EcoRV (Invitrogen); (2) gDNA of populations 16S and 45R was digested with three restriction endonucleases, XbaI (Invitrogen), KpnI (Invitrogen) and BamHI (NEB); (3) populations 16S, 161S, 45R and 46R gDNA digestion was repeated with HindIII, to confirm the results obtained in the first two experiments.

The genomic DNA (30 μg per sample) was digested overnight at 37 °C in a total volume of 400 μL, following manufacturer’s instructions for each restriction enzyme. Aliquots of the digestion were checked in 1% agarose gel and the digestion stopped when the gDNA was completely digested. DNA was precipitated and re-suspended in 30 μL of ddH_2_O. Digested DNA and an undigested control were run on a 0.8% agarose gel in TBE buffer overnight using the DNA molecular weight market III DIG-labeled (Roche) for reference.

The gel was immersed in HCl 0.25 M stirring for 15 min at room temperature, followed by 20 min in denaturation buffer (0.5 M NaOH, 1.5 M NaCl). It was then washed in neutralization buffer (0.5 M -Tris-HCl pH 7.5, 1.5 M NaCl) for 30 min. DNA on gel was transferred to positively charged nylon membrane (Roche) by capillary in a 20× SSC buffer (3 M NaCl, 0.3 M sodium citrate, pH 7 and autoclaved) overnight. On the following day, DNA was fixed on the membrane using the Stratalinker^®^ UV Crosslinker with the appropriate program and the hybridization was carried out following the manufacturer’s instructions. DIG-labeled probe was hybridized at 49 °C overnight in DIG Easy Hyb solution (Roche) to detect the homologous DNA fragments on the DNA blot.

### 2.4. ALS Gene Sequencing and Cloning

Two cloning experiments, with different aims, were conducted. In the first one, the full-length sequences of the *ALS* gene of two plants of each population were cloned, sequenced and aligned to detect the SNPs and mutations endowing resistance. In the second cloning experiment, the aim was to identify the diverse *ALS* gene copies of each species and their phylogenetic relationships. As it is technically difficult to clone a 2 kbp amplicon, in this experiment the *ALS* gene was amplified and cloned in two parts: 620 bp starting from the 5′ of the gene and including the aa position 122, and 1000 bp ending at the 3′ of the gene and including the aa position 574.

Total RNA was extracted from 100 mg of one fresh young leaf using the commercial InviTrap Spin Plant RNA Mini Kit (Invitek, Berlin, Germany). cDNA was synthesized using the ImProm-II^TM^ Reverse Transcriptase System (Promega) following the manufacturer’s instructions and the different parts of the *ALS* gene were amplified using the primer pairs reported in Table 1. PCR amplifications were conducted using the Advantage 2 PCR Kit (Clontech, Takara) in a 50 µL mixture of 1× Advantage 2 SA PCR Buffer, 1× dNTP mix (10 mM each), 0.2 µM of each primer, 1× Advantage 2 Polymerase Mix and 100 ng cDNA. Amplifications were conducted using the following program: 1 min at 95 °C; 35 cycles of 30 s at 95 °C, 30 s at 60 °C, and 40–120 s (depending on the length of the amplicon) at 68 °C; 3 min at 68 °C. PCR products were purified through columns using MinElute PCR purification kit (Qiagen) and cloned using the TOPO TA Cloning^®^ kit (Invitrogen) following the manufacturer’s instructions. In the first experiment, six colonies in total were selected from two plants per population. Plasmids were extracted from *E. coli* bacterial cells using the PureYieldTM Plasmid Miniprep System (Promega), quantified in 1% agarose gel and sequenced with universal M13 primers and a forward primer positioned in the middle of the gene (ECH_5F). In the second experiment, 10–20 colonies were selected for each of two plants per population and only the universal M13 primers were used for the sequencing. The nucleotide sequences were edited using DNASTAR^®^ software and phylogenetic analysis was conducted using MEGA X^®^ software to determine specific motifs in the sequences able to distinguish the homologous gene copies. The analysis was hierarchically performed: in the first step cloned sequences of a single plant were analyzed together: then, those of different plants belonging to the same population: then, those of populations belonging to the same species and then all together.

### 2.5. Relative Expression of ALS Gene Copies 

This study was used to analyze the relative expression of the *ALS* gene copy carrying the mutations responsible for target-site resistance to the ALS inhibitor penoxsulam (*ALS1*) with respect to the expression of the other *ALS* gene copies (*ALS2-3*). Comparisons were also made among populations belonging to different *Echinochloa* species, having different resistance pattern and collected in sites far away from each other.

Two S populations (16S and 161S) and three R populations (44R, 45R and 46R) were included in the analyses. Twenty seedlings for each population were transplanted into pots (one plant per pot) containing a soil substrate and maintained in a greenhouse until they reached the three-leaf stage. Ten plants for each population were sprayed with penoxsulam 240 g L^−1^ at the dose of 250 mL ha^−1^. After 24 h, 50–100 mg of leaf tissue was sampled from three treated (T) and three non-treated (NT) plants of each population. Total RNA was extracted using the TRIzol method [27], purified using the DNase I (Roche) and cDNA was synthesized using the SuperScript^®^ III Reverse Transcriptase (Invitrogen) following the manufacturer’s instructions.

For qPCR analyses, specific primers for the different *ALS* gene copies were designed based on an *ALS* gene part including two allele-specific SNPs (Table 2). The resulting amplicon was 140 bp long and included a cutting site for the restriction endonuclease FokI only in *ALS1.* Therefore, a preliminary check was performed on cDNA samples of different *Echinochloa* populations to check the specificity of each primer pair.

Eight reference genes (*Rubisco*, *Actin*, *18S*, *CAP*, *Tubuline*, *GAPDH*, *EF1* and *Ubiquitine*) chosen among those previously tested on *Echinochloa* spp. [28] were tested on the cDNA of two populations (161S and 45R). Serial dilutions of cDNA template (1:100, 1:50, 1:25 and 1:6.25) were tested to set up the appropriate experimental conditions. Finally, cDNA of T and NT plants of all populations included in the experiment was tested in the qPCR analyses to determine the relative expression of the different *ALS* gene copies using two reference genes (*Rubisco* and *18S*). qPCR was carried out in three technical replicates using the SYBR Green^®^ kit in a 7300 Real-Time PCR System (Applied Biosystems, Corning, NY, USA) on 96-well plates PCR-96M2-HS-C^®^ (Axygen, Corning, NY, USA) with a sealer MicroAmp^®^ Optical Adhesive Film (Applied Biosystems, Corning, NY, USA). The reactions were performed in a final volume of 20 μL consisting of 10 μL of cDNA (diluted 1:25), 2 μL of 10× buffer, 1.2 μL of MgCl_2_ solution (50 mM), 0.5 μL of dNTPs (10 μM of each nucleotide), 2 μL of SYBR Green^®^ (Invitrogen), 0.2 μL of ROX Reference Dye (Invitrogen), 0.1 μL of Taq Platinum^®^ (Invitrogen) and 0.4 μL of each of the forward and reverse primers (10 μM). The amplification steps included an initial cycle of 95 °C for 5 min, followed by 40 cycles including a three-step amplification sequence (94 °C for 15 s, 60 °C for 10 s, 72 °C for 15 s), a fourth step where the instrument detects the fluorescence emitted at each cycle (60 °C for 35 s), and a final elongation step of 95 °C for 15 s and 60 °C for 60 s.

Data were analyzed using the Sequence Detection Software (SDS) 1.4 and the Ct values means, the standard deviation and confidence interval per treatment were calculated. The relative expression was calculated on the mean Ct values using the ∆Ct method [29] by the following equation:(1)$$\Delta \Delta Ct=\left(Ct_{target}-Ct_{reference}\right)-(Ct_{calibrator}-Ct_{reference})$$
where ∆∆*Ct* is the relative expression of the different *ALS* gene copies in our case, and the application of the result in 2^−(^^∆∆*Ct*)^ gives the variation dimension (Relative Quantity, RQ). The Ct reference values were determined from the average of the two reference genes considered.
genes-12-01841-t002_Table 2Table 2List of primers used for the study of relative expression of the different *ALS* gene copies.Primer NamePrimer Sequence (5′–3′)TargetF1_590TmGAG CAC ACA CAT ACT TGG GGC AT ^1^Forward primer, pos. 590Specific for *ALS1*F2_590CmAGC ACA CAC ATA CTT GGG GCA C ^1^Forward primer, pos. 590Specific for *ALS2-3*R1_621GAG CAT CTT CTT AAT TGC TGC ACG GReverse primer, pos. 621Specific for *ALS1*R2_621GAG CAT CTT CTT GAT TGC TGC ACG TReverse primer, pos. 621Specific for *ALS2-3*^1^ Underlined, a further mismatch was introduced to improve the discrimination of the primer [30].


## 3. Results

### 3.1. Echinochloa Classification and Herbicide Resistance Patterns 

The classification of the populations collected from fields based on the different classification keys led to the assignment of three populations to the “red” genus *E. crus-galli* (16S, 95R and 100R) due predominantly to the red color of the stems’ base and the small mean size of the seeds (Appendix A). Regarding the other four populations, using the Pignatti classification key, only population 161S could be assigned to *E. erecta* species. Due to the heterogeneity of plants’ traits, the other three populations (44R, 45R and 46R) could only be assigned to the “white” group. Considering the Pignatti classification key [17], they could be assigned to *E. erecta* and *E. phyllopogon* based on the absence or presence, in some plants but not in others, of hairs at the green stems’ base. According to the Costea and Tardif classification key [18], they could be assigned to *E. oryzicola* species based on the large mean size of the seeds (Appendix A). Finally, considering the Tabacchi and Viggiani classification key [19], they could be assigned to the *E. phyllopogon* or *E. oryzicola* genus based on the presence/absence of hairs and to the folding and colors of the leaves.

Herbicide resistance assays showed that two populations, 16S and 161S, were completely controlled by all herbicides tested (i.e., at least two ALS inhibitors and one ACCase inhibitor) (Appendix A), whereas the other five populations, 44R, 45R, 46R, 95R and 100R, showed a high level of resistance to at least one ALS inhibitor (Appendix A): populations 45R, 46R and 95R were highly cross-resistant to imazamox, azimsulfuron and penoxsulam; population 44R was highly resistant to penoxsulam and imazamox, but controlled by azimsulfuron; population 100R was only tested for resistance to penoxsulam and showed high levels of resistance.

### 3.2. DNA Barcoding and CAPS-rbcL

In a preliminary study [24], some typical DNA barcoding genes were analyzed on a series of previously identified *Echinochloa* spp. accessions (each accession was a bulk of seeds derived from a single mother plant). The sequences of *rbcL* obtained from two plants (A and B in Figure 1) of each of seven accessions and from nine vouchered sequences of *rbcL* from the database of the NCBI GenBank nucleotide were selected and aligned. Phylogenetic analysis highlighted a clear division between two groups of species, the so called “red” (*E. crus-galli*) and “white” (including *E. phyllopogon*, *E. erecta* and *E. oryzicola*) (Figure 1). The division obtained by the analyses of *rbcL* DNA-barcoding confirmed the results obtained using the Tabacchi and Viggiani classification key, i.e., all species belonging to the “white” group may be classified as *E. oryzicola*.

Amplification with the generic primers CAPS-rbcL-F and CAPS-rbcL-R produced a single band of 217 bp (Appendix A), which is cut when subjected to a digestion with the endonuclease TasI in two bands of 134 and 87 bp only in plants belonging to a “white” species (Appendix A). Among plants analyzed with the marker CAPS-rbcL for the validation of the method (42 samples in total), it was confirmed that three populations included only plants belonging to “red” species, i.e., *E. crus-galli* (16S, 95R and 100R), and four populations included only plants belonging to “white” species, i.e., *E. oryzicola* (161S, 44R, 45R and 46R), confirming the classification conducted using the dichotomous keys (Figure 2).

### 3.3. Southern Blotting with ALS Gene

The DIG-labeled DNA probe of 360 bp corresponding to a conserved region including region B of the *ALS* gene (corresponding to amino acid positions 573–576) was used to detect the homologous fragments on genomic DNA blots. This led to quantifying the number of *ALS* loci in different *Echinochloa* species’ genomes with different herbicide resistance patterns. The experiments gave variable results depending on the endonuclease used: results of hybridization of the DNA fragments digested with HindIII in the first (Appendix A) and third experiments (Figure 3) and with XbaI (Appendix A) in the second one revealed that one more copy of the *ALS* gene was present in the *E. crus-galli* populations (three) compared with the *E. oryzicola* populations (two). Furthermore, it seems that the number of *ALS* gene copies was not linked to the resistant status of the populations: the three *E. oryzicola* plants had the same profile when digested with HindIII, with two *ALS* gene copies identified (Figure 3). The visible band at 21,226 bp was not considered because it corresponds to undigested genomic DNA. 

### 3.4. ALS Gene Sequencing and Cloning

Analysis of the alignment of cDNA full-length *ALS* cloned sequences identified many SNPs (data not shown). They may be associated with the presence of homologous copies of the gene that could be subjected to natural modification during evolution. The SNPs detected that are associated with known mutations responsible for ALS inhibitors resistance [31] were analyzed individually. Two such mutations were found: in position 122 a double point mutation GG-AA, giving the amino acid change Ala-Asn, was detected in the resistant populations 95R and 100R of *E. crus-galli*, and at position 574, a point mutation G-T, giving the amino acid change Trp-Leu, was found in the resistant plants of *E. oryzicola* populations 44R, 45R and 46R.

Phylogenetic analysis of the full-length cloned sequences allowed the homologous gene copies to be distinguished in the different species. The first step of the analyses included the clones of single plants: an example cladogram including six selected clones of the 16S-1 plant and six clones of the 45R-4 plant (Figure 4A) showed that three clusters are distinguishable: *ALS1* includes clones of both plants (the three clones from pop 45R carry the point mutation Trp574Leu), *ALS2* includes only sequences of the 45R population (which did not carry the point mutation at 574) and *ALS3* includes only clones of the population 16S. If more full-length cloned sequences of the *E. crus-galli* 16S population are considered (Figure 5A), they are distributed across all three *ALS* gene clusters. These results are in line with the results obtained with the Southern blotting analyses, which identified two *ALS* copies for *E. oryzicola* and three for *E. crus-galli* (Figure 3).

Analyses of the partial sequences of the *ALS* gene clones identified two and three *ALS* gene copies for populations belonging to the *E. oryzicola* and *E. crus-galli* species, respectively. Figure 4B reports the cladogram obtained from the alignment and phylogenetic analyses of 79 partial sequences obtained from the amplification of the 3′ part of the *ALS* gene of plants belonging to the *E. oryzicola* populations 161S, 44R and 45R. Meanmhile, Figure 5B reports a cladogram that includes 46 partial sequences obtained from the amplification of the 5′ part of the *ALS* gene of plants belonging to the *E. crus-galli* populations 16S, 95R and 100R. To identify the different gene copies in the cladogram of the partial sequences, the full-length sequences of population 45R for *E. oryzicola* reported in Figure 4A and those of population 16S for *E. crus-galli* reported in Figure 5A were used. To identify in which *ALS* gene copy the mutations 122 and 574 were present, only the 3′-end and 5′-end of the full-length sequences were aligned with the partial sequences of the second cloning experiment. Comparing the clustering of the selected sequences in Figure 4A,B (for *E. oryzicola*) and in Figure 5A,B (for *E. crus-galli*), it was possible to conclude that the *ALS1* gene copy is the one carrying the mutation in both species (for clarity corresponding clones present in both cladograms are highlighted; furthermore, in the cladogram of *E. crus-galli* 5′-end sequences, clones including the 122 mutation are highlighted because the correspondence of full-length and partial sequences was achieved using the sequences of susceptible population 16S).

Confirmation of the clustering of *ALS* homologous genes was obtained with the alignment of 138 cloned partial sequences of the 3′-end of the *ALS* gene for all populations included in the analyses (Appendix A). Three homologs were identified, 55% of the sequences clustered in *ALS1*, 30% in *ALS2* and 15% in *ALS3*. In *ALS3,* only sequences from *E. crus-galli* populations are present, and all mutated clones from both species clustered in *ALS1*.

### 3.5. Relative Expression of ALS Gene Copies

The sequence analyses identified several SNPs among the different gene copies that can be used to distinguish *ALS1* from *ALS2*-3 and for the design of allele-specific primers for the *ALS* gene copy expression analysis (Table 2). Two SNPs in positions 590 (T/C) and 621 (C/A) of the aa sequences of the *ALS* gene were detected to specifically identify *ALS1* (the copy of the gene that carries the mutations responsible for target-site resistance) from the other homologs (*ALS2-3*).

Among the eight reference genes analyzed, the two with higher expression, on average, higher and with more primer specificity were chosen as reference genes for the gene expression study, i.e., *Rubisco* and *18S* (an example of amplification plot and dissociation curves obtained for these two genes is reported in Appendix A, whereas all graphs were stored in a dedicated repository [32]). From these reference genes, a robust expression profile for the quantitative expression was obtained of the gene of interest (*ALS*) and, in particular, of the relative expression of the different *ALS* gene copies (*ALS1* and *ALS2-3*). The Ct values of the two reference genes were very similar for all populations (Appendix A), from 19.6 to 24.9 for *Rubisco* and from 20.4 to 23.7 for *18S*, and in both treated and untreated plants. Therefore, the mean values of the two reference genes were used for the calculation of the relative expression of the *ALS* alleles.

The specificity of the primers designed for *ALS1* and *ALS2-3* was confirmed by the endonuclease analyses performed with FokI (Appendix A): only the amplicon obtained from cDNA samples amplified with the primers F1_590Tm+R1_621 (Table 2) was cut to produce a band of 105 bp, whereas the same cDNA samples amplified with the primers F2_590Cm+R2_621 produced the expected amplicon of 140 bp, which was not cut by FokI. A combination of the two bands was never observed, confirming the specificity of the primer pairs.

Amplification plots obtained from the qPCR analyses of *ALS* genes with the specific primers indicated that all samples had high template levels, i.e., curves with Ct from 23 to 34 (an example of amplification plot and dissociation curves obtained for *ALS1* and *ALS2-3* is reported in Appendix A), whereas graphs for all other populations tested were stored in a dedicated repository [32]. On the right of the amplification plot, under the threshold line (green), it is possible to see the amplification of the non-template control (checks including only water). Dissociation curves showed that non-specific products were not amplified by the F1_590Tm + R1_621 primer pair, whereas some secondary peaks are visible in the graph of *ALS2-3*, most likely indicating some formation of primer dimers. The problem was limited to a few plant replicates, which were removed from the analyses.

The ∆*Ct* method allows us to relatively quantify (RQ) the expression of genes of interest (*ALS1* and *ALS2-3* in our case) with respect to reference genes (*Rubisco* and *18S*) choosing as a base the expression of a sample called the “calibrator” (in our case the susceptible non-treated). With two susceptible populations in the experiment (16S and 161S), data were elaborated using the untreated plants of both populations as the calibrator. Ct data showed that the two susceptible populations are slightly different (Figure 6): Ct values were comparable if treated samples were considered (28.3 and 27.6 for population 16S and 161S, respectively) and no differences were reported between *ALS1* and *ALS 2-3*, whereas Ct values for non-treated samples were higher for population 161S. For non-treated samples of both S populations, Ct values were slightly lower for *ALS1* compared to *ALS2-3*, indicating that the expression of *ALS1* was higher (Figure 6).

Relative expression data of the different *ALS* gene copies are reported in Figure 7 for the untreated (Figure 7A) and herbicide treated plants (Figure 7B). The untreated plants showed base level expression for the different alleles, suggesting that in non-stressed plants, *ALS1* is, on average, more expressed than *ALS2-3*. In the treated samples, these differences were highlighted also in the susceptible populations (Figure 7B).

## 4. Discussion

In this study we considered a series of *Echinochloa* spp. populations collected in rice fields from different areas of Italy with a broad high resistance level to ALS inhibitors. In fact, both phenotypes associated with Ala122Asn and Trp574Leu mutations showed, on average, high resistance levels to all ALS inhibitors tested: penoxsulam, azimsulfuron and imazamox (note that population 100R was tested only for resistance to penoxsulam, therefore no speculation can be made regarding the other ALS inhibitors). An exception was population 44R which showed a slightly different pattern, it was cross-resistant to imazamox and penoxsulam but controlled by azimsulfuron. Furthermore, population 44R was one of the first cases of *Echionochloa* resistant to the ACCase inhibitor profoxydim detected in Italy [11]. The broad cross-resistance pattern was previously found in other species for mutations in position 122 [33] and also in *Echinochloa* spp. carrying a mutation in position 574 [11,12], whereas the resistance pattern of population 44R was firstly associated with the mutation in position 574 [31].

The aim of this study was to clearly identify the main *Echinochloa* species involved in ALS inhibitors resistance in Italian rice fields and to characterize the associated target-site resistance mechanism. Previous studies identified mutations on the *ALS* gene associated with herbicide resistance, but information about the effect of mutations on the different genomes of this polyploid genus is not available. The morphological observations made according to different classification keys and the molecular data obtained analyzing the *rbcL* gene of several *Echinochloa* populations indicate that the two prevalent *Echinochloa* species infesting Italian rice fields are *E. crus-galli* and *E. oryzicola*. This conclusion is supported also by the studies on ITS and cpDNA conducted by Aoki and Hirofumi on several *Echinochloa* spp. accessions collected in Japan [34]. They demonstrated that *E. phyllopogon* and *E. oryzicola* belong to the same species (i.e., similar morphological, cytogenetic, and isozymic features, as well as cross-compatibility) and, therefore, *E. phyllopogon* is a synonym of *E. oryzicola*.

DNA-blotting analysis revealed differences in the *ALS* gene locus number in the different populations investigated. The results of the hybridization of DNA fragments cut with HindIII showed that two copies of the *ALS* gene are present in the genome of the *E. oryzicola* susceptible population 161S and resistant populations 45R and 46R, whereas one more copy was detected in the genome of the *E. crus-galli* susceptible population 16S. These results were confirmed by the *ALS* gene cloning analyses, which revealed the presence of two shared homologs in all species (*ALS1* and *ALS2*) to which a third copy (*ALS3*) was added when *E. crus-galli* populations were included as described also by Runge et al. [35]. The differences in the gene copy numbers are in accordance with the genetic background of *E. crus-galli* (6n), which is assumed to be evolutionarily derived from interspecific crossing between tetraploid *E. oryzicola* (4n) and an unidentified diploid *Echinochloa* spp. (2n) [16]. The additional *ALS* gene whose orthologue is not in *E. oryzicola* is likely from the genome of the unidentified diploid *Echinochloa* spp.

*ALS1* was demonstrated to be the gene copy that carries the mutations responsible for target-site resistance. In a study on *E. crus-galli* [36] it was asserted that all three *ALS* gene copies identified are involved in the evolution of target-site resistance, even if the results showed that mainly one *ALS* gene copy (named *ALS3*) was involved in conferring resistance. This second finding is in line with our qPCR results which identified *ALS1* as the gene copy, on average, that was more expressed in both non-treated and herbicide treated samples. Another study on a different weed, *Monochoria vaginalis*, identified five copies of *ALS* genes, two of which contributed to the evolution of target-site resistance and were largely more abundant than the other *ALS* gene copies [37].

It seems that in *Echinochloa* spp. the higher expression of *ALS1* was not linked neither to the species, nor to the resistance status or the origin of the populations. *ALS1* was expressed more highly in both population 16S, which belongs to *E. crus-galli* and in all other populations that belong *E. oryzicola*. At the same time, *ALS1* was expressed more highly in both susceptible, 16S and 161S, and resistant populations 44R, 45R and 46R, which are originated from different parts of Italy. The relatively higher expression of *ALS1* was expected considering the outcome of the cloning of the *ALS* gene. It was observed that, at least for *E. oryzicola*, *ALS1* clones were preferentially selected in the random sampling over *ALS2* clones.

The identification of resistance-endowing mutations only in *ALS1* is not entirely unexpected [38], since this copy was also found to be the most highly expressed. In a polyploid species with multiple *ALS* copies, a dilution effect will occur if only one of the copies carries resistance [15,39,40]. The less the resistance-conferring gene is expressed, relative to the other copies, the greater the dilution will be, resulting in a lower level of resistance. Instead, in *Avena fatua* [39] single *ACCase* resistance mutations confer relatively low resistance levels, likely due to a dilution effect by susceptible *ACCase* expressed by homoeologs which led to a slow resistance evolution. The lack of dilution effect observed in *Echinochloa* spp. [36] may be associated with the rapid evolution of resistance seen in recent years in these species.

Although NTSR mechanisms cannot be ruled out [10,41], TS *ALS1* mutations seem to be the most evolutionarily favored mechanism endowing ALS inhibitor resistance in *Echinochloa* spp. It would be interesting to conduct a broader survey of *Echinochloa* spp. to determine whether resistance-conferring mutations in the other *ALS* copies are occasionally present and, if so, to quantify the level of whole-plant resistance that they confer. Other questions that would be interesting to answer are: which ancestral genome does *ALS1* belong to? In the case of multiple-resistance, do different resistance-endowing variations occur on the same ancestral genome?

## Figures and Tables

**Figure 1 genes-12-01841-f001:**
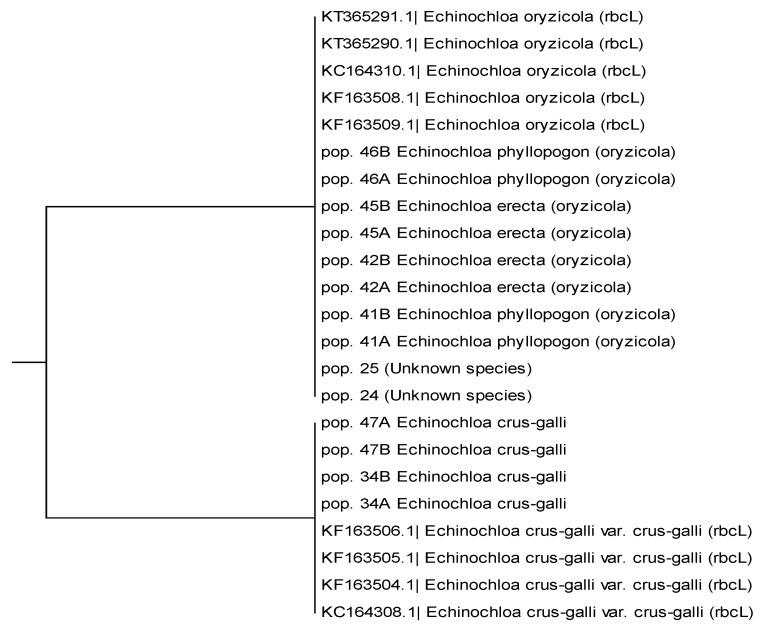
Cladogram obtained from UPMGA estimation of nine *Echinochloa rbcL* sequences from GenBank database (identified by GenBank ID, e.g., KT365291.1) and 14 sequences from seven previously identified accessions (e.g., 41A and 41B are two plants derived from seeds of the same mother plant) harvested in Italian rice fields.

**Figure 2 genes-12-01841-f002:**
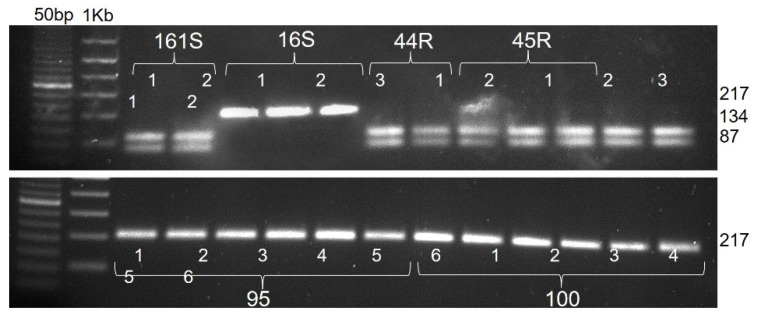
Results of CAPS-rbcL molecular marker for a selection of samples tested: plants belonging to “white” species, *E. oryzicola* (161S, 44R, 45R and 46R), show the two digested bands at 134 and 87 bp, whereas plants belonging to “red” species, *E. crus-galli* (16S, 95R and 100R) show a single band.

**Figure 3 genes-12-01841-f003:**
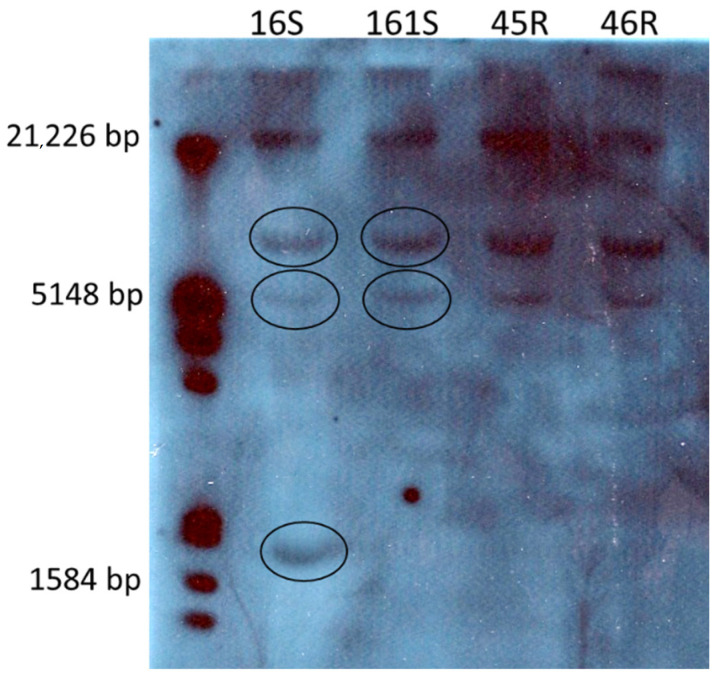
Southern blotting analysis on *ALS* gene of *E. crus-galli* (16S) and *E. oryzicola* (161S, 45R and 46R) genomic DNA extracted from leaf tissue. DNA samples of different populations (reported on the top) were digested with the enzyme HindIII. The numbers on the left show the DIG-labeled molecular ladder. Circles indicate the copies of the *ALS* gene in the different species.

**Figure 4 genes-12-01841-f004:**
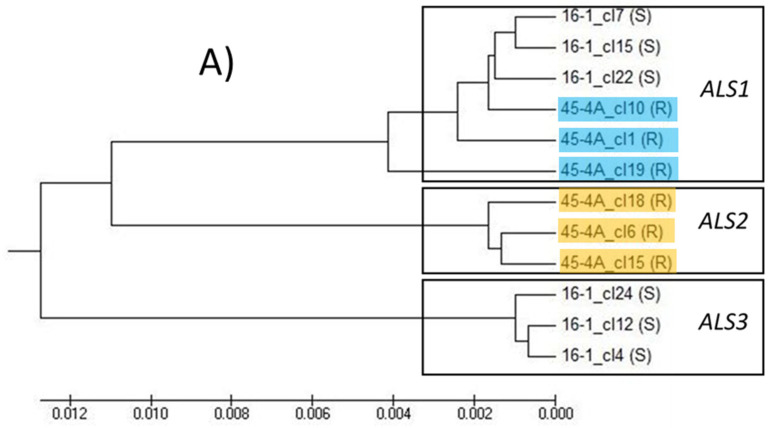
(**A**) Cladogram obtained from UPMGA estimation of twelve *ALS* full-length clone sequences, six of the S sample 16-1 (population 16S plant 1) and six of the R sample 45-4A (population 45R plant 4A), using MEGA X^®^; the mutation Trp574Leu was detected in all the sequences of the *ALS1* of R samples; (**B**) cladogram obtained from UPMGA estimation of 79 *ALS* partial clone sequences starting from 3′-end belonging to the *E. oryzicola* populations 161S, 44R and 45R, using MEGA X^®^. The corresponding sequences in *ALS1* and *ALS2* are highlighted in blue and orange, respectively. Sequences in blue of population 45R carry the mutation 574, other sequences mutated are present in *ALS1* but are not highlighted to avoid confusion.

**Figure 5 genes-12-01841-f005:**
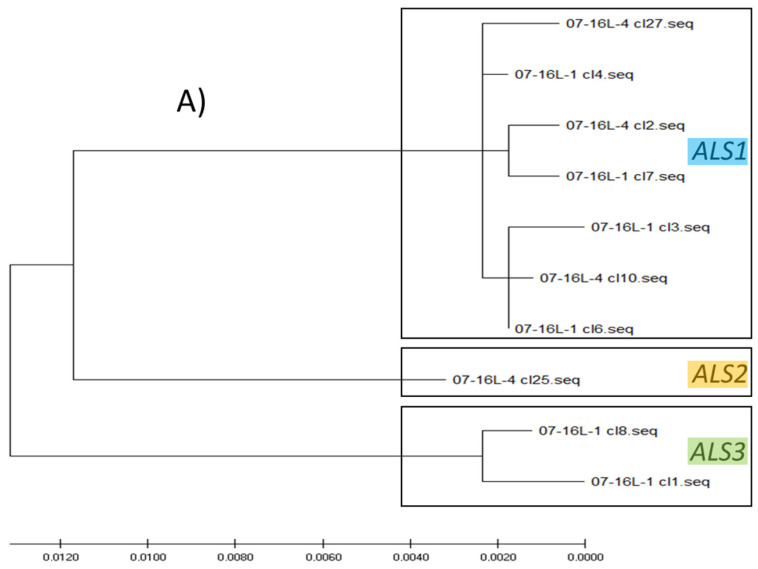
(**A**) Cladogram obtained from UPMGA estimation of full-length clone sequences, five of the plant 16L-1 and five of the plant 16L-4 (both plants of S *E. crus-galli* population 16S) using MEGA X^®^; these clones were used to define the different *ALS* gene copies in the R populations of the *E. crus-galli* species; (**B**) cladogram obtained from UPMGA estimation of 46 *ALS* partial clone sequences starting from 5′-end belonging the *E. crus-galli* populations 16S, 95R and 100R, using MEGA X^®^; the mutated clones of *E. crus-galli* populations 95R and 100R are highlighted in red; the sequences of population 16S present in the cladogram of (**A**) are highlighted in blue (*ALS1*), orange (*ALS2*) and green (*ALS3*), respectively, in the cladogram of (**B**).

**Figure 6 genes-12-01841-f006:**
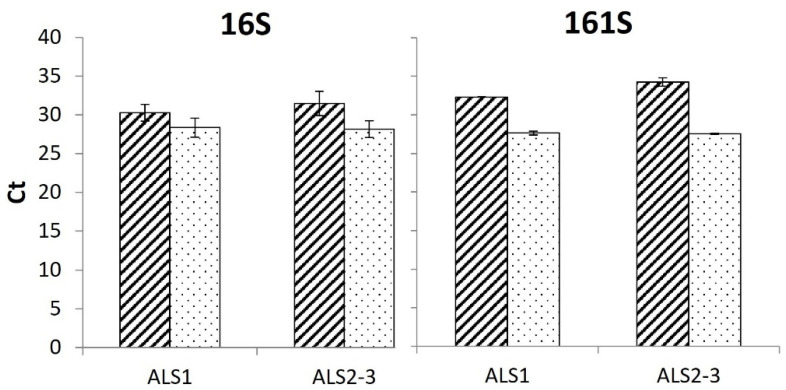
Ct of *ALS1* and *ALS2-3* amplified on the susceptible populations, 16S and 161S. Hatched bars indicate the mean values of the untreated plants, whereas dotted bars indicate the mean values of the treated plants. Thin bars represent the standard errors.

**Figure 7 genes-12-01841-f007:**
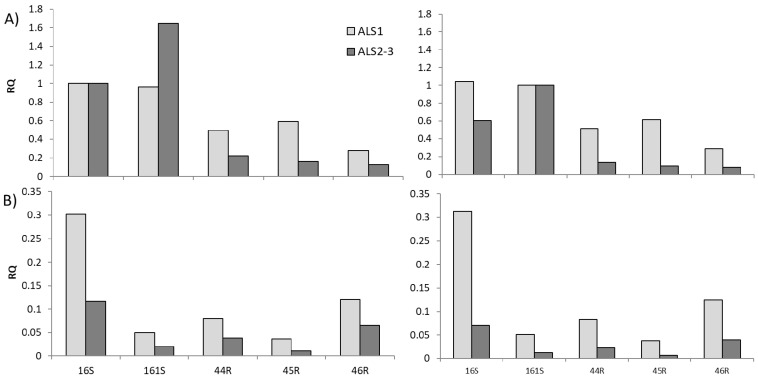
Relative expression (RQ) of the *ALS* gene copies (*ALS1* light grey bars and *ALS2-3* dark grey bars) in the (**A**) untreated plants and (**B**) treated plants. The mean data of the two reference genes, *Rubisco* and *18S*, were considered and the untreated control of populations 16S and 161S on the left and right graphs, respectively, were used as calibrator.

**Table 1 genes-12-01841-t001:** List of primers used to amplify different parts of the *ALS* gene and their target.

Primer Name	Primer Sequence (5′–3′)	Target
ECH_F	TCG CAA GGG CGC GGA CAT CCT CGT	Forward primer, cloning full length and 5′ part of the gene
5RACE-1	GCC GCG ACT CAC CAA CAA GA	Reverse primer, cloning 5′ part of the gene
ECH_5F	AGG TCA CSC GCT CCA TCA CCA	Forward primer, cloning 3′ part of the gene
ECH-3R	TCC TGC CAT CAC CHT CCA KGA	Reverse primer, cloning full length and 3′ part of the gene

## Data Availability

The data presented in this study are openly available in “qPCR data of *ALS* gene copies expression in *Echinochloa* spp.” at [32].

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
