# Peer review of "Target-Site Mutations and Expression of ALS Gene Copies Vary According to Echinochloa Species"

_genes, 2021, doi:10.3390/genes12111841_

Round 1

Reviewer 1 Report

In this study, the authors investigated the ALS herbicide resistance mechanism in Echinochloa spp. found in Italy. They firstly classified Echinochloa spp. into species with morphology and molecular markers. The classified species were analyzed for ALS gene mutation, a major resistance mechanism to this class of herbicides. The authors found that the target-site mutations only in ALS1 and concluded that the bias is caused by the expression level of ALS1.

Echinochloa spp. is a noxious weed of rice cultivation and its resistance has been a problem in many parts of the world. However, molecular analysis of Echinochloa spp. resistance has not been sufficiently carried out because of the difficulty of genetic analysis due to the problem of ploidy. Thus, molecular study in the weeds may provide an important insight. In addition, the effect of duplicated genes on the evolution of resistance, which is the main proposition of this study, is an important topic in herbicide-resistance study. The effects of gene duplication in target-site genes on the expression and/or evolution of herbicide resistance have not been clearly answered in weed resistance research, although studies at the enzymatic and molecular levels have revealed several important findings.

Unfortunately, the entire level of experiments and analyses is not sufficient to meet the magnitude of the addressed task. Two major flaws are observed in gene sequencing and real-time PCR.

Cloning experiments are always fraught with the problem of PCR errors. The PCR enzyme they used is not specified, but if it is the same as the one used for CAPS analysis, it is an error-prone enzyme, and it is not appropriate to use it to distinguish between homeologs. It is necessary to use high fidelity PCR enzymes or to amplify and analyze each homeolog separately by PCR. In fact, these methods have been used in the analysis of Echinochloa spp. In addition to fidelity, cloning experiment is known to cause recombination mediated by PCR, which may lead to misinterpretation of the results. For example, chimeric sequence between ALS1 and other ALS genes can be generated, and can be cloned in the study. In the whole paper, there is no detailed description of the method, and if the analysis takes these into account to the maximum extent, it should be described. The paper does not refer to the details of the methodologies, so it is not clear how accurate the experiments are. If the analysis takes these into account, it has to be clearly described.

The real-time PCR method also has a major problem. In this study, real-time PCR was used to compare the expression levels of different genes. This is the analysis that real-time PCR is not good at and should not be used without a lot of cares. While the amount of template in the cDNA definitely affects the signal intensity, the length of the amplicon also has a significant effect in the SYBR green method. Namely, the same CT value does not guarantee the same template amount in the cDNA. Also, even if the signal intensity per molecule of the amplicons is the same, the comparison cannot be made unless the amplification efficiency of the genes is exactly the same. This study attempts to detect small differences in expression levels. In this case, even a small difference in amplification efficiency of a few percent can have a large impact on the results. The only way to compare expression between genes by real-time PCR is to use the absolute quantification method, which is also not very accurate unfortunately. Since the quantification of gene expression affects the very basis of this research, it is desirable that it be done very carefully.

Finally, although not mentioned at all in the paper, another group reported that all the ALS genes are used in the evolution of ALS inhibitor resistance in E. crus-galli (Löbmann et al. 2021 Journal of Plant Diseases and Protection). The lack of at least a mention of this paper and a discussion of the discrepancies between the studies may reflect the lack of effort the authors put into understanding the background of their research. A similar, more careful study has also been reported recently (Tanigki et al. 2021 New Phytologist). I recommend that the research be re-engaged in light of these analyses as well.

Author Response

We thank the reviewer for his/her comments and hope to have addressed all the requested modifies. See attached file.

Reviewer 2 Report

This is a nicely-designed study to clarify resistance to ALS inhibitors in Echinochloa spp. in Italy. But of course, there are always things that can be improved!

Figure 6: It is unclear why this figure is included. If you find it important to include this data, perhaps move it to the supplemental figures.

Figure 7: Untreated plants should be presented in the left bar and treated plants on the right. Was a change in expression expected after herbicide treatment? This point is not fully described in the text. And, why were 20 plants grown, with 10 plants treated and 10 plants not treated (and not even mock-treated), if only six plants were used for expression analysis? How were the plants selected for analysis?

Figure 8: I don't understand what the main message of this figure is. Is it how expression changes after herbicide treatment for all three gene copies? Why are ALS2-3 combined? Is the same data shown in the left and right plots but with different "calibrator" lines? The differences in scale between the top and bottom plots make it hard to compare expression across all four plots. Why were only five lines (16S, 161S, 44R, 45R, 46R) included and not 95R and 100R? These two lines, 95R and 100R, are also barely mentioned in the discussion.

The discussion of the higher expression of ALS1 (521-526) is not easy to follow. I cannot remember all the details required to understand how the statements in parentheses help explain anything. I suggest breaking this section up into a few more sentences - devoting a little more space to the topic may help clarify it.

I did not quite follow the description of primer design in the methods (91-96). A preliminary analysis was done but what did it accomplish? Was a definitive analysis also performed?

Population 100R was only tested with penoxsulam which limits the conclusions you can draw - why not test it with the other ALS-inhibitors? Then you can draw complete conclusions for this study.

I recommend trying to reduce the number of times you use the phrase "due to". 

Author Response

(The authors gave the same response as above.)

Reviewer 3 Report

General comments to authors

Authors investigated the ALS genes involved in target-site resistance of polyploid Echinochloa populations present in Italian rice fields. In particular, using a CAPS-rbcL molecular marker, the E. crus-galli (L.) P. Beauv. and E. oryzicola (Vasinger) Vasing. species were recognized as the most common species in rice fields, whereas the molecular characterization led to identification of three ALS genes in E. crus-galli and two in E. oryzicola. In addition, the two species carried different point mutations conferring resistance: Ala122Asn in E. crus-galli and Trp574Leu in E. oryzicola. The mutations were carried in the ALS1 gene copy, which was more expressed than the ALS2 and ALS3 copies in both species. Although these findings are very interesting from a scientific point of view, there are some flaws throughout the manuscript which should be taken into consideration in order to improve its quality.

Specific comments to authors

Please explain the reason for not testing azimsulfuron activity against the E. oryzicola 161S and E. crus-galli 100R populations.

I suggest that Figure 3 should be replaced by a better quality image.

Line 488, it should be mentioned that the E. oryzicola 44R was cross-resistant to imazamox and penoxsulam, susceptible to azimsulfuron and partially resistant to profoxydim.

Line 490, the above resistance pattern of the E. oryzicola 44R is reported for first time.

Lines 519-529, Please make comments about the fact that the ALS1 was demonstrated to be the only gene copy that carries the mutations responsible for target-site resistance and to be more expressed in both non-treated and herbicide treated samples, and in both weed species either susceptible or resistant. In addition, please compare your findings to those related to the hexaploid wild oat (Avena fatua) species with resistance to ACCase inhibiting herbicides (Yu Q, MS Ahmad-Hamdani, H Han, MJ Christoffers and SB Powles. 2013. Herbicide resistance-endowing ACCase gene mutations in hexaploid wild oat (Avena fatua): insights into resistance evolution in a hexaploid species. Heredity 110, 220–231)

Author Response

(The authors gave the same response as above.)

Round 2

Reviewer 1 Report

The focal point of the paper

If the paper focuses on just the detection of point mutation, the authors should remove all the data regarding expression because the part makes the quality of the paper worse. However, the very points raised by the authors such as difference in expression levels among the ALS gene copies, frequency of TSR mutation represent the study of the effect of duplicated genes on the evolution of resistance.

The problem of cloning and sequencing experiment

The PCR enzyme used in this study is not classified as “high fidelity” enzyme. Taq-based enzyme is PCR error-prone, to a lesser or greater degree. You will never see double peaks in the cloning experiments even if chimeric clones exist. I guess the authors misunderstand the generation of chimera by PCR or they do not understand cloning and sequencing experiments? Just in case, here are two examples describing the chimera generation by PCR.

Bradey, R.D. and D.M.Hillis. 1997. Recombinant DNA Sequence Generated by PCR Amplification. Molecular Biology and Evolution 14:592-593.

Cronn,R., M.Cedoni, T.Haselkorn, and C,Grover. 2002. PCR-mediated recombination in amplification products derived from polyploid cotton. Theoretical and Applied Genetics 104:482-289.

>The same technique to define the gene copies of ACCase gene was used by Yu et al. (2013, Heredity 110, 220–231).

What other researchers do is not a guarantee of correctness. Every experimental technology has its weaknesses. The question is (1) whether or not the weaknesses are understood by the researchers and (2) whether or not they devise ways to prevent those weaknesses from becoming a problem, and (3) whether or not they conduct verification by other experimental technologies. So far, the authors did not provide enough answers to any of the three points.

It is very hard to understand the experimental procedures performed in this study (section 3.3). The authors have to understand that when cloning experiments cannot be avoided, multiple clones need to be sequenced for each allele, which has to be subjected to consensus analysis. Without the consensus analysis, polymorphisms between individuals cannot be discussed (L342). This analysis is costly as many clones need to be sequenced. So, it is common to amplify each gene for direct sequencing.

One of the paper the authors cited,  Iwakami et al. (2015) seems to analyze the ALS sequence by direct sequencing. Or NGS can be applied as performed in hexaploid Echinochloa (McElroy and Hall 2020).

I cannot understand why the authors believe the results are consistent with the Southern blot analysis (L361). The figure shows 16(S) plant carries three alleles for ALS1 and ALS3… The above mentioned paper, Iwakami et al. 2015, performed a similar analysis, where the ALS sequences were compared between the two species. The authors should include the previously published data for their analysis, whether or not to include it in the paper.

I cannot understand the difference of Fig. 4 and 5.

The problem of qPCR-based gene-expression comparisons BETWEEN genes

As mentioned in my comments for the 1st submission, Ct values are affected by multiple factors such as length of PCR products, PCR efficiency, relative quantification cannot be applied to compare gene expression between genes. If the authors want to compare gene expression between the genes, use the absolute quantification method of qPCR, or other experimental technologies. I need to apologize that I misunderstood at least about the size of the PCR products. Still, the problem of amplification efficiency is not addressed. Also, I noticed the problem of internal control data (see below). I would rather remove all the expression data that are from scientifically unsound experiments, although then the manuscript would be hard to be accepted in Genes, which, if I understand correctly, is for “a broad audience”.

The guideline for amplification efficiency in ddCT is often 90-110%. So, I speculate the efficiency of the primers tested by the authors is within the range. Let’ say, the amplification efficiency is 1.90-fold for gene A and 1.95-fold for gene B, and just a single copy of the cDNA molecule for each gene exists in the PCR template. At the time of 30 cycles, the products from gene A and gene B becomes 230466617.9 and 502386939.9 copies, respectively, with an approximately 2.17-fold difference. In the case of 1.90-fold and 2.00-fold, the difference of the product copies becomes 4.65-fold. Thus, ddCT method should not be used in situations where a difference as small as this can be a problem.

I did not notice at the 1st submission, but there is a huge variation in the expression of internal control genes. This indicates the genes cannot be used as internal controls. Better genes must be used.

Discussion regarding the exclusivity of ALS1 as TSR gene

Unfortunately, it is very difficult to understand the logic being developed here. On one hand, the authors say the expression level is not related to the ALS locus for TSR evolution (L530). On the other hand, the authors say the expression has an influence on it (L539, L549).